# Mean Field for the Stochastic Blockmodel: Optimization Landscape and Convergence Issues

**Soumendu Sunder Mukherjee**[*]
Interdisciplinary Statistical Research Unit (ISRU)
Indian Statistical Institute, Kolkata
Kolkata 700108, India
soumendu041@gmail.com

**Purnamrita Sarkar**[*]
Department of Statistics and Data Science
University of Texas, Austin
Austin, TX 78712
purna.sarkar@austin.utexas.edu

**Y. X. Rachel Wang**[*]
School of Mathematics and Statistics
University of Sydney
NSW 2006, Australia
rachel.wang@sydney.edu.au

**Bowei Yan**
Department of Statistics and Data Science
University of Texas, Austin
Austin, TX 78712
boweiy@utexas.edu

## Abstract

Variational approximation has been widely used in large-scale Bayesian inference recently, the simplest kind of which involves imposing a mean field assumption to approximate complicated latent structures. Despite the computational scalability of mean field, theoretical studies of its loss function surface and the convergence behavior of iterative updates for optimizing the loss are far from complete. In this paper, we focus on the problem of community detection for a simple two-class Stochastic Blockmodel (SBM). Using batch co-ordinate ascent (BCAVI) for updates, we show different convergence behavior with respect to different initializations. When the parameters are known, we show that a random initialization can converge to the ground truth, whereas in the case when the parameters themselves need to be estimated, a random initialization will converge to an uninformative local optimum.

## 1 Introduction

Variational approximation has recently gained a huge momentum in contemporary Bayesian statistics [13, 5, 11]. Mean field is the simplest type of variational approximation, and is a popular tool in large scale Bayesian inference. It is particularly useful for problems which involve complicated latent structure, so that direct computation with the likelihood is not feasible. The main idea of variational approximation is to obtain a tractable lower bound on the complete log-likelihood of any model. This is, in fact, akin to the Expectation Maximization algorithm [6], where one obtains a lower bound on the marginal log-likelihood function via the expectation with respect to the conditional distribution of the latent variables under the current estimates of the underlying parameters. In contrast, for mean field variational approximation, the lower bound or ELBO is computed using the expectation with respect to a product distribution over the latent variables.

While there are many advances in developing new mean field type approximation methods for Bayesian models, the theoretical behavior of these algorithms is not well understood. There is one line of work that studies the asymptotic consistency of variational inference. Most of the existing theoretical work focuses on the global optimizer of variational methods. For example, for Latent

---
[*]Equal contribution.

Dirichlet Allocation (LDA) [5] and Gaussian mixture models, it is shown in [16] that the global optimizer is statistically consistent. [23] connects variational estimators to profile M-estimation, and shows consistency and asymptotic normality of those estimators. For Stochastic Blockmodels (SBM) [10, 9], [3] shows that the global optimizer of the variational log-likelihood is consistent and asymptotically normal. For more general cases, [22] proves a variational Bernstein-von Mises theorem, which states that the variational posterior converges to "the Kullback-Leibler minimizer of a normal distribution, centered at the truth".

Recently, a lot more effort is being directed towards understanding the statistical convergence behavior of non-convex algorithms in general. For Gaussian mixture models and exponential families with missing data, [19, 21] prove local convergence to the true parameters. The same authors also show that the covariance matrix from variational Bayesian approximation for the Gaussian mixture model is "too small" compared with that obtained for the maximum likelihood estimator [20]. The robustness of variational Bayes estimators is further discussed in [8]. For LDA, [2] shows that, with proper initialization, variational inference algorithms converge to the global optimum.

To be concrete, let us take the community detection problem in networks. Here the latent structure involves unknown community memberships. Optimization of the likelihood involves a combinatorial search, and thus is infeasible for large-scale graphs. The mean field approximation has been used popularly for this task [4, 26]. In [3], it was proved that the global optimum of the mean field approximation to the likelihood behaves optimally in the dense degree regime, where the average expected degree of the network grows faster than the logarithm of the number of vertices.

In [26], it is shown that if the initialization of mean field is close enough to the truth then one gets convergence to the truth at the minimax rate. However, in practice, it is usually not possible to initialize like that unless one uses a pilot algorithm. Most initialization techniques like Spectral Clustering [17, 15] will return correct clustering in the dense degree regime, thus rendering the need for mean field updates redundant.

Indeed, in most practical scenarios, one simply uses multiple random initializations, which usually fails miserably. However, to understand the behavior of random initializations, one needs to better understand the landscape of the mean field loss. There are few such works for non-convex optimization in the literature; notable examples include [14, 7, 12, 24]. In [24], the authors fully characterize the landscape of the likelihood of the equal proportion Gaussian Mixture Model with two components, where the main message is that most random initializations should indeed converge to the ground truth. In contrast, for topic models, it has been established that, for some parameter regimes, variational inference exhibits instability and returns a posterior mean that is uncorrelated with the truth [7]. In this respect, for network models, there has not been much work characterizing the behavior of the variational loss surface.

In this article, in the context of a stochastic blockmodel, we give a complete characterization of all the critical points and establish the behavior of random initializations for batch co-ordinate ascent (BCAVI) updates for mean field likelihood (with known and unknown model parameters). Our results thus complement the results of [25].

For simplicity, we work with equal-sized two class stochastic blockmodels. We show that, when the model parameters are known, random initializations can converge to the ground truth. We also analyze the setting with unknown model parameters, where they are estimated jointly with the cluster memberships. In this case, we see that indeed, with high probability, a random initialization never converges to the ground truth, thus showing the critical importance of a good initialization for network models.

## 2   Setup and preliminaries

The stochastic blockmodel $\texttt{SBM}(B, Z, \pi)$ is a generative model of networks with community structure on $n$ nodes. Its dynamics is as follows: there are $K$ communities $\{1, \ldots, K\}$ and each node belongs to a single community, where this membership is captured by the rows of the $n \times K$ matrix $Z$, where the $i$th row of $Z$, i.e. $Z_{i\star}$, is the community membership vector of the $i$th node and has a $\texttt{Multinomial}(1; \pi)$ distribution, independently of the other rows. Given the community structure, links between pairs of nodes are determined solely by the block memberships of the nodes in an independent manner. That is, if $A$ denotes the adjacency matrix of the network, then given $Z$, $A_{ij}$

and $A_{kl}$ are independent for $(i,j) \neq (k,l)$, $i < j$, $k < l$, and
$$\mathbb{P}(A_{ij} = 1 \mid Z) = \mathbb{P}(A_{ij} = 1 \mid Z_{ia} = 1, Z_{jb} = 1) = B_{ab}.$$
$B = ((B_{ab}))$ is called the block (or community) probability matrix. We have the natural restriction that $B$ is symmetric for undirected networks.

The block memberships are hidden variables and one only observes the network in practice. The goal often is to fit an appropriate SBM to learn the community structure, if any, and also estimate the parameters $B$ and $\pi$.

The complete likelihood for the SBM is given by
$$\mathbb{P}(A, Z; B, \pi) = \prod_{i<j} \prod_{a,b} (B_{ab}^{A_{ij}}(1 - B_{ab})^{1-A_{ij}})^{Z_{ia}Z_{jb}} \prod_i \prod_a \pi_a^{Z_{ia}}. \tag{1}$$
As $Z$ is not observable, if we integrate out $Z$, we get the data likelihood
$$\mathbb{P}(A; B, \pi) = \sum_{Z \in \mathcal{Z}} \mathbb{P}(A, Z; B, \pi), \tag{2}$$
where $\mathcal{Z}$ is the space of all $n \times K$ matrices with exactly one 1 in each row.

In principle we can optimize the data likelihood to estimate $B$ and $\pi$. However, $\mathbb{P}(A; B, \pi)$ involves a sum over a complicated large finite set (the cardinality of this set is $K^n$), and hence is not easy to deal with. A well-known alternative approach is to optimize the variational log-likelihood [3], which has a less complicated dependency structure, the simplest of which is mean field log-likelihood (see, e.g., [18]). We defer a detailed discussion of the mean field principle in the supplementary material.

For the SBM, the variational log-likelihood with respect to a distribution $\psi$ is given by
$$\sum_Z \log \left( \frac{\mathbb{P}(A, Z; B, \pi)}{\psi(Z)} \right) \psi(Z) = \mathbb{E}_\psi \left( \sum_{i<j,a,b} Z_{ia}Z_{jb}(\theta_{ab}A_{ij} - f(\theta_{ab})) \right) - \mathtt{KL}(\psi || \pi^{\otimes n}),$$
where $\theta_{ab} = \log \left( \frac{B_{ab}}{1-B_{ab}} \right)$, $f(\theta) = \log(1 + e^\theta)$ and $\pi^{\otimes n}$ denotes the product measure on $\mathcal{Z}$ with the rows of $Z$ being i.i.d. $\mathtt{Multinomial}(1; \pi)$. A special case of the variational log-likelihood is the mean field log-likelihood (see, e.g., [18]), where one approximates $\Psi$ by
$$\Psi_{MF} \equiv \{\psi : \psi(z_1, \ldots, z_n) = \prod_{j=1}^n \psi_j(z_j)\}. \tag{3}$$
Define $\ell_{MF}(\psi, \theta, \pi) = \sum_{i<j,a,b} \psi_{ia}\psi_{jb}(\theta_{ab}A_{ij} - f(\theta_{ab})) - \sum_i \mathtt{KL}(\psi_i || \pi)$. For SBM the mean field approximation is equivalent to optimizing $\ell_{MF}(\psi, \theta, \pi)$ as follows:
$$\max_\psi \ell_{MF}(\psi, \theta, \pi)$$
$$\text{subject to} \sum_a \psi_{ia} = 1, \text{ for all } 1 \leq i \leq n$$
$$\psi_{ia} \geq 0, \text{ for all } 1 \leq i \leq n, 1 \leq a \leq K,$$
where each $\psi_i$ is a discrete probability distribution over $\{1, \ldots, K\}$.

## 2.1 Mean field updates for a two-parameter two-block SBM

Consider the stochastic blockmodel with two blocks with prior block probability $\pi, 1 - \pi$ respectively and block probability matrix $B = (p-q)I + qJ$, where $p > q$, $I$ is the identity matrix, and $J = \mathbf{1}\mathbf{1}^\top$ is the matrix of all 1's. For simplicity, we will denote $\psi_{i1}$ as $\psi_i$. Then the mean field log-likelihood is
$$\ell(\psi, p, q, \pi) = \frac{1}{2} \sum_{i,j:i\neq j} [\psi_i(1 - \psi_j) + \psi_j(1 - \psi_i)][A_{ij} \log \left( \frac{q}{1-q} \right) + \log(1-q)]$$
$$+ \frac{1}{2} \sum_{i,j:i\neq j} [\psi_i\psi_j + (1 - \psi_i)(1 - \psi_j)][A_{ij} \log \left( \frac{p}{1-p} \right) + \log(1-p)]$$
$$- \sum_i [\log \left( \frac{\psi_i}{\pi} \right) \psi_i + \log \left( \frac{1-\psi_i}{1-\pi} \right) (1 - \psi_i)].$$

For simplicity of exposition, we will assume that $\pi$ (which is essentially a prior on the block memberships) is known and equals $1/2$. Let $\mathcal{C}_i, i = 1, 2$ be the two communities. Let $\tilde{\pi} = \frac{|\mathcal{C}_1|}{n}$. It is clear that $\tilde{\pi} = \frac{1}{2} + O_P(\frac{1}{\sqrt{n}})$. Assuming $\tilde{\pi} = \frac{1}{2}$ from the start will not change our conclusions but make the algebra a lot nicer, which we do henceforth. Now

$$
\begin{aligned}
\frac{\partial \ell}{\partial \psi_i} &= \frac{1}{2} \sum_{j:j\neq i} 2[1 - 2\psi_j][A_{ij} \log \left( \frac{q}{1-q} \right) + \log(1-q)] \\
&\quad + \frac{1}{2} \sum_{j:j\neq i} 2[2\psi_j - 1][A_{ij} \log \left( \frac{p}{1-p} \right) + \log(1-p)] - \log \left( \frac{\psi_i}{1-\psi_i} \right) \\
&= 4t \sum_{j:j\neq i} (\psi_j - \frac{1}{2})(A_{ij} - \lambda) - \log \left( \frac{\psi_i}{1-\psi_i} \right),
\end{aligned}
$$

where $t = \frac{1}{2} \log \left( \frac{p(1-q)}{q(1-p)} \right)$ and $\lambda = \frac{1}{2t} \log \left( \frac{1-q}{1-p} \right)$. Detailed calculations of other first and second order partial derivatives are given in Section 2 of the supplementary article [1]. The co-ordinate ascent (CAVI) updates for $\psi$ are

$$
\log \frac{\psi_i^{(new)}}{1 - \psi_i^{(new)}} = 4t \sum_{j\neq i} (\psi_j - \frac{1}{2})(A_{ij} - \lambda).
$$

Introducing an intermediate variable $\xi$ for the updates, let $f(x) = \log(\frac{x}{1-x})$ and $\xi_i = f(\psi_i)$. Then at iteration $s$, the batch version (BCAVI) of this is

$$
\xi^{(s)} = 4t(A - \lambda(J - I))(\psi^{(s-1)} - \frac{1}{2}\mathbf{1}),
$$

and $\psi^{(s)} = g(\xi^{(s)})$ with $g(x) = 1/(1 + e^{-x})$. The population version (replacing $A$ by $\mathbb{E}(A \mid Z) = ZBZ^\top - pI =: P - pI$) of BCAVI is

$$
\xi^{(s)} = 4t(P - pI - \lambda(J - I))(\psi^{(s-1)} - \frac{1}{2}\mathbf{1}).
$$

The matrix $M := P - pI - \lambda(J - I)$ will appear many times later. There are updates for $p, q$ as well, which can be expressed compactly in terms of $\psi$. We describe these in detail in (8).

## 3 Main results

In this section, we state and discuss our main results. All the proofs appear in the supplementary article [1].

**Note:** In the following, we will see the following vectors repeatedly: $\psi = \frac{1}{2}\mathbf{1}, \mathbf{1}, \mathbf{0}, \mathbf{1}_{\mathcal{C}_1}, \mathbf{1}_{\mathcal{C}_2}$. Among these, $\mathbf{1}$ corresponds to the case where every node is assigned by $\psi$ to $\mathcal{C}_1$, and, similarly, for $\mathbf{0}$, to $\mathcal{C}_2$. On the other hand, $\mathbf{1}_{\mathcal{C}_i}$ are the indicators of the clusters $\mathcal{C}_i$ and hence correspond to the ground truth community assignment. Finally, $\frac{1}{2}\mathbf{1}$ corresponds to the solution where a node belong to each community with equal probability.

**Proposition 3.1.** *Suppose $1 > p > q > 0$. Then*

1. $\frac{(p-q)(1+p-q)}{2(1-q)p} < t < \frac{(p-q)(1-p+q)}{2(1-p)q}$, *and*

2. $q < \lambda < p$.

The eigendecomposition of $P - \lambda J$ will play a crucial role in our analysis. Note that it has rank two and two eigenvalues $e_\pm = n\alpha_\pm$, where $\alpha_+ = \frac{p+q}{2} - \lambda, \alpha_- = \frac{p-q}{2}$, with eigenvectors $\mathbf{1}$ and $\mathbf{1}_{\mathcal{C}_1} - \mathbf{1}_{\mathcal{C}_2}$ respectively.

Now, the eigenvalues of $M$ are $\nu_1 = e_+ - (p - \lambda)$, $\nu_2 = e_- - (p - \lambda)$ and $\nu_j = -(p - \lambda)$, $j = 3, \ldots, n$. The eigenvector of $M$ corresponding to $\nu_1$ is $u_1 = \mathbf{1}$, and the one corresponding to $\nu_2$ is $u_2 = \mathbf{1}_{\mathcal{C}_1} - \mathbf{1}_{\mathcal{C}_2}$.

### 3.1 Known $p, q$:

In this case, we need only consider the updates for $\psi$. The population BCAVI updates are

$$\xi^{(s+1)} = 4tM(\psi^{(s)} - \frac{1}{2}\mathbf{1}). \tag{4}$$

We consider the case where the true $p$, $q$ are of the same order, that is, $p \asymp q \asymp \rho_n$ with $\rho_n$ possibly going to 0. In the known $p, q$ case $\frac{1}{2}\mathbf{1}$ is a saddle point of the population mean field log-likelihood.

**Proposition 3.2.** $\psi = \frac{1}{2}\mathbf{1}$ *is a saddle point of the population mean field log-likelihood when $p$ and $q$ are known, for all $n$ large enough.*

Now we will write the BCAVI updates in the eigenvector coordinates of $M$. To this end, define $\zeta_i^{(s)} = \langle \psi^{(s)}, u_i \rangle / \|u_i\|^2 = \langle \psi^{(s)}, u_i \rangle / n$, for $i = 1, 2$. We can then write

$$\psi^{(s)} = \langle \psi^{(s)}, u_1/\|u_1\| \rangle u_1/\|u_1\| + \langle \psi^{(s)}, u_2/\|u_2\| \rangle u_2/\|u_2\| + v^{(s)} = \zeta_1^{(s)} u_1 + \zeta_2^{(s)} u_2 + v^{(s)}.$$

So, using (4) in conjunction with the above decomposition, coordinate-wise we have:

$$\xi_i^{(s+1)} = 4tn\left((\zeta_1^{(s)} - \frac{1}{2})\alpha_+ + \sigma_i \zeta_2^{(s)} \alpha_-\right) + 4t\nu_3\left((\zeta_1^{(s)} - \frac{1}{2}) + \sigma_i \zeta_2^{(s)} + v_i^{(s)}\right) \tag{5}$$

$$=: na_{\sigma_i}^{(s)} + b_i^{(s)}, \tag{6}$$

where $\sigma_i = 1$, if $i$ is in $\mathcal{C}_1$, and $-1$ otherwise.

**Theorem 3.3** (Population behavior). *The limit behavior of the population BCAVI updates is characterized by the signs of $\alpha_+$ and $a_{\pm 1}^{(0)}$, where $\alpha_+ = (p + q)/2 - \lambda$ and $a_{\pm 1}^{(s)}$ for iteration $s$ is defined in (5). Assume that $|na_{\pm 1}^{(0)}| \to \infty$. Define $\ell(\psi^{(0)}) = \mathbb{1}(a_{+1}^{(0)} > 0)\mathbf{1}_{\mathcal{C}_1} + \mathbb{1}(a_{-1}^{(0)} > 0)\mathbf{1}_{\mathcal{C}_2}$. Then, we have*

$$\frac{\|\psi^{(1)} - \ell(\psi^{(0)})\|^2}{n} = O(\exp(-\Theta(n\min\{|a_{+1}^{(0)}|, |a_{-1}^{(0)}|\}))) = o(1).$$

*We also have for any $s \geq 2$*

$$\frac{\|\psi^{(s)} - \ell(\psi^{(0)})\|^2}{n} = \begin{cases} O(\exp(-\Theta(nt\alpha_-))), & \text{if } a_{+1}^{(0)} a_{-1}^{(0)} < 0, \\ O(\exp(-\Theta(nt|\alpha_+|))), & \text{if } a_{+1}^{(0)} a_{-1}^{(0)} > 0, \text{ and } \alpha_+ > 0. \end{cases}$$

*Finally, if $a_{+1}^{(0)} a_{-1}^{(0)} > 0$ and $\alpha_+ < 0$, then, for any $s \geq 2$, we have*

$$\min\left\{\frac{\|\psi^{(s)} - \mathbf{1}\|^2}{n}, \frac{\|\psi^{(s)} - \mathbf{0}\|^2}{n}\right\} = O(\exp(-\Theta(nt|\alpha_+|))).$$

*In fact, in this case, $\psi^{(s)}$ cycles between $\mathbf{1}$ and $\mathbf{0}$, in the sense that it is close to $\mathbf{1}$ is one iteration, and to $\mathbf{0}$ in the next and so on.*

**Remark 3.1.** *We see from Theorem 3.3 that, essentially, we have exponential convergence within two iterations.*

Now we turn to the sample behavior. To distinguish from the population case, we denote the sample BCAVI updates as

$$\hat{\xi}^{(s+1)} = 4t\hat{M}(\hat{\psi}^{(s)} - \frac{1}{2}\mathbf{1}), \tag{7}$$

where $\hat{M} = A - \lambda(J - I)$ and $\hat{\psi}^{(s)}$ depends on $A$ for $s \geq 1$. Note that $\hat{\psi}^{(0)} = \psi^{(0)}$.

**Theorem 3.4** (Sample behavior). *For all $s \geq 1$, the same conclusion as Theorem 3.3 holds for the sample BCAVI updates with high probability as long as $n|a_{\pm 1}^{(0)}| \gg \max\{\sqrt{n\rho_n}\|\psi^{(0)} - \frac{1}{2}\|_\infty, 1\}$, $\sqrt{n\rho_n} = \Omega(\log n)$ and $\psi^{(0)}$ is independent of $A$.*

From Theorem 3.3, we can calculate lower bounds to the volumes of the basins of attractions of the limit points of the population BCAVI updates. We have the following corollary.

**Corollary 3.5.** *Define the set of initialization points converging to a stationary point* $\mathbf{c}$ *as*

$$\mathcal{S}_{\mathbf{c}} := \{v \mid \limsup_{s \to \infty} n^{-1} \|\psi^{(s)} - \mathbf{c}\|^2 = O(\exp(-\Theta(nt \min\{|\alpha_+|, \alpha_-\}))), \text{ when } \psi^{(0)} = v\}.$$

*Let* $\mathfrak{M}$ *be some measure on* $[0, 1]^n$, *absolutely continuous with respect to the Lebesgue measure. Consider the stationary point* $\mathbf{1}$, *then*

$$\mathfrak{M}(\mathcal{S}_{\mathbf{1}}) \geq \lim_{\gamma \uparrow 1} \mathfrak{M}(H_+^\gamma \cap H_-^\gamma \cap [0, 1]^n),$$

*where the half-spaces* $H_\pm^\gamma$ *are given as*

$$H_\pm^\gamma = \Big\{ x \mid \langle x, \alpha_+ u_1 \pm \alpha_- u_2 \rangle > \frac{n\alpha_+}{2} + \frac{n^{1-\gamma}}{4t} \Big\}.$$

*Similar formulas can be obtained for the other stationary points.*

For specific measures $\mathfrak{M}$, one can obtain explicit formulas for these volumes. In practice, these are quite easy to calculate by Monte Carlo simulations.

In fact, using arguments that goes into the proof of Theorem 3.3, we can show that in the large $n$ limit, there are only five stationary points of the mean field log-likelihood, namely $\frac{1}{2}\mathbf{1}, \mathbf{1}, \mathbf{0}, \mathbf{1}_{\mathcal{C}_1}$, and $\mathbf{1}_{\mathcal{C}_2}$.

### 3.2  Unknown $p, q$:

In this case, the BCAVI updates are

$$p^{(s)} = \frac{(\psi^{(s-1)})^\top A \psi^{(s-1)} + (1 - \psi^{(s-1)})^\top A (1 - \psi^{(s-1)})}{(\psi^{(s-1)})^\top (J - I) \psi^{(s-1)} + (1 - \psi^{(s-1)})^\top (J - I)(1 - \psi^{(s-1)})}, \tag{8}$$

$$q^{(s)} = \frac{(\psi^{(s-1)})^\top A (1 - \psi^{(s-1)})}{(\psi^{(s-1)})^\top (J - I)(1 - \psi^{(s-1)})}, \tag{9}$$

$$t^{(s)} = \frac{1}{2} \log \left( \frac{p^{(s)}(1 - q^{(s)})}{q^{(s)}(1 - p^{(s)})} \right), \quad \lambda^{(s)} = \frac{1}{2t^{(s)}} \log \left( \frac{1 - q^{(s)}}{1 - p^{(s)}} \right),$$

$$\xi^{(s)} = 4t^{(s)}(A - \lambda^{(s)}(J - I))(\psi^{(s-1)} - \frac{1}{2}\mathbf{1}).$$

Similar to before, $p \asymp q \asymp \rho_n$ with $\rho_n$ possibly going to 0. In the population version, we would replace $A$ with $\mathbb{E}(A \mid Z) = P - pI$.

In this case with unknown $p, q$, our next result shows that $\frac{1}{2}\mathbf{1}$ changes from a saddle point (Proposition 3.2) to a local maximum.

**Proposition 3.6.** *Let* $n \geq 2$. *Then* $(\psi, p, q) = (\frac{1}{2}\mathbf{1}, \frac{\mathbf{1}^\top A \mathbf{1}}{n(n-1)}, \frac{\mathbf{1}^\top A \mathbf{1}}{n(n-1)})$ *is a strict local maximum of the mean field log-likelihood.*

Since $p, q$ and $\psi$ are unknown and need to be estimated iteratively, we have the following updates for $p^{(1)}$ and $q^{(1)}$ given the initialization $\psi^{(0)}$ and show that they can be written in terms of the projection of the initialization in the principal eigenspace of $P$.

**Lemma 3.1.** *Let* $x = \psi^T \psi + (1 - \psi)^T (1 - \psi)$ *and* $y = 2\psi^T (1 - \psi) = n - x$. *If* $\psi = \zeta_1 u_1 + \zeta_2 u_2 + w$, *where* $w \in \text{span}\{u_1, u_2\}^\perp$, *then*

$$p^{(1)} = \frac{p + q}{2} + \frac{(p - q)(\zeta_2^2 - x/2n^2)}{\zeta_1^2 + (1 - \zeta_1)^2 - x/n^2} + O_P(\sqrt{\rho_n}/n),$$

$$q^{(1)} = \frac{p + q}{2} - \frac{(p - q)(\zeta_2^2 + y/2n^2)}{2\zeta_1(1 - \zeta_1) - y/n^2} + O_P(\sqrt{\rho_n}/n). \tag{10}$$

Since $\psi^T (1 - \psi) > 0$, we have $\zeta_1(1 - \zeta_1) \geq \zeta_2^2$. This gives:

$$p^{(1)} \in \left( \frac{p + q}{2} + O_P(\sqrt{\rho_n}/n), p \right], \qquad q^{(1)} \in \left[ q, \frac{p + q}{2} + O_P(\sqrt{\rho_n}/n) \right). \tag{11}$$

It is interesting to note that $p^{(1)}$ is always smaller than $q^{(1)}$ except when it is $O(\sqrt{\rho_n}/n)$ close to $(p+q)/2$. In that regime, one needs to worry about the sign of $t$ and $\lambda$. In all other regimes, $t, \lambda$ are positive.

Using the update forms in Lemma 3.1, the following result shows that the stationary points of the population mean field log-likelihood lie in the principle eigenspace $\text{span}\{u_1, u_2\}$ of $P$ in a limiting sense.

**Proposition 3.7.** *Consider the case with unknown p, q and $\rho_n \to 0$, $n\rho_n \to \infty$. Let $(\psi, \tilde{p}, \tilde{q})$ be a stationary point of the population mean field log-likelihood. If $\psi = \psi_u + \psi_{u^\perp}$, where $\psi_u \in \text{span}\{u_1, u_2\}$ and $\psi_{u^\perp} \perp \text{span}\{u_1, u_2\}$, then $\|\psi_{u^\perp}\| = o(\sqrt{n})$ as $n \to \infty$.*

Lemma 3.1 basically shows that if $\zeta_2$ is vanishing, then $p^{(1)}$ and $q^{(1)}$ concentrates around the average of the conditional expectation matrix, i.e. $(p+q)/2$. The next result shows that if one uses independent and identically distributed initialization, then $\zeta_2$ is indeed vanishing. This is not surprising, since $\zeta_2$ measures correlation with the second eigenvector of $P$ $u_2$ which is basically the $\mathbf{1}_{\mathcal{C}_1} - \mathbf{1}_{\mathcal{C}_2}$ vector.

Consider a simple random initialization, where the entries of $\psi^{(0)}$ are i.i.d with mean $\mu$ and show that it converges to $\frac{1}{2}$ with small deviations within one update. This shows the futility of random initialization.

**Lemma 3.2.** *Consider the initial distribution $\psi_i^{(0)} \overset{iid}{\sim} f_\mu$ where $f$ is a distribution supported on $(0, 1)$ with mean $\mu$. If $\mu$ is bounded away from 0 and 1 and $n\rho_n = \Omega(\log^2 n)$, then $\psi_i^{(1)} = \frac{1}{2} + O_P(\log n/\sqrt{n})$ uniformly for all i, where $\psi^{(1)}$ is computed using the sample update.*

Perhaps, it is also instructive to analyze the case where the initialization is in fact correlated with the truth, i.e. $E[\psi_i^{(0)}] = \mu_{\sigma_i}$. To this end, we will consider the following initialization scheme.

**Lemma 3.3.** *Consider an initial $\psi^{(0)}$ such that*

$$\zeta_1 = \frac{(\psi^{(0)})^T \mathbf{1}}{n} = \frac{\mu_1 + \mu_2}{2} + O_P(1/\sqrt{n}),$$

$$\zeta_2 = \frac{(\psi^{(0)})^T u_2}{n} = \frac{\mu_1 - \mu_2}{2} + O_P(1/\sqrt{n}). \tag{12}$$

*If $\mu_1, \mu_2$ are bounded away from 0 and 1 and satisfy*

$$|\mu_1 - \mu_2| > \max\left(2|\mu_1 + \mu_2 - 1| + O_P\left(\frac{\sqrt{\rho_n \log^2 n/n}}{p-q}\right), \left(\frac{\rho_n \log n}{n(p-q)^2}\right)^{1/3}\right), \tag{13}$$

*and $n\rho_n = \Omega(\log^2 n)$, then $\psi^{(1)} = \mathbf{1}_{\mathcal{C}_1} + O_P(\exp(-\Omega(\log n)))$ or $\mathbf{1}_{\mathcal{C}_2} + O_P(\exp(-\Omega(\log n)))$, where the error term is uniform for all the coordinates.*

**Remark 3.2.** *The lemma states that provided the separation between p and q does not vanish too fast, if the initial $\psi^{(0)}$ is centered around two slightly different means, e.g., $\mu_1 = 1/2 + c_n$ and $\mu_2 = 1/2 - c_n$ for some constant $c_n \to 0$, then we converge to the truth within one iteration.*

## 4 Numerical results

In Figure 1-(a), we have generated a network from an SBM with parameters $p = 0.4, q = 0.025$, and two equal sized blocks of 100 nodes each. We generate 5000 initializations $\psi^{(0)}$ from $\text{Beta}(\alpha, \beta)^{\otimes n}$ (for four sets of $\alpha$ and $\beta$) and map them to $a_{\pm 1}^{(0)}$. We perform sample BCAVI updates on $\psi^{(0)}$ with known $p, q$ and color the points in the $a_{\pm 1}^{(0)}$ co-ordinates according the limit points they have converged to. In this case, $\alpha_+ > 0$, hence based on Theorems 3.3 and 3.4, we expect points with $a_{+1}^{(0)} a_{-1}^{(0)} < 0$ to converge to the ground truth (colored green or magenta) and those with $a_{+1}^{(0)} a_{-1}^{(0)} > 0$ to converge to $\mathbf{0}$ or $\mathbf{1}$. As expected, points falling in the center of the first and third quadrants have converged to $\mathbf{0}$ or $\mathbf{1}$. The points converging to the ground truth lie more toward the boundaries but mostly remain in the same quadrants, suggesting possible perturbations arising from the sample noise and small network size. We see that this issue is alleviated when we increase $n$.

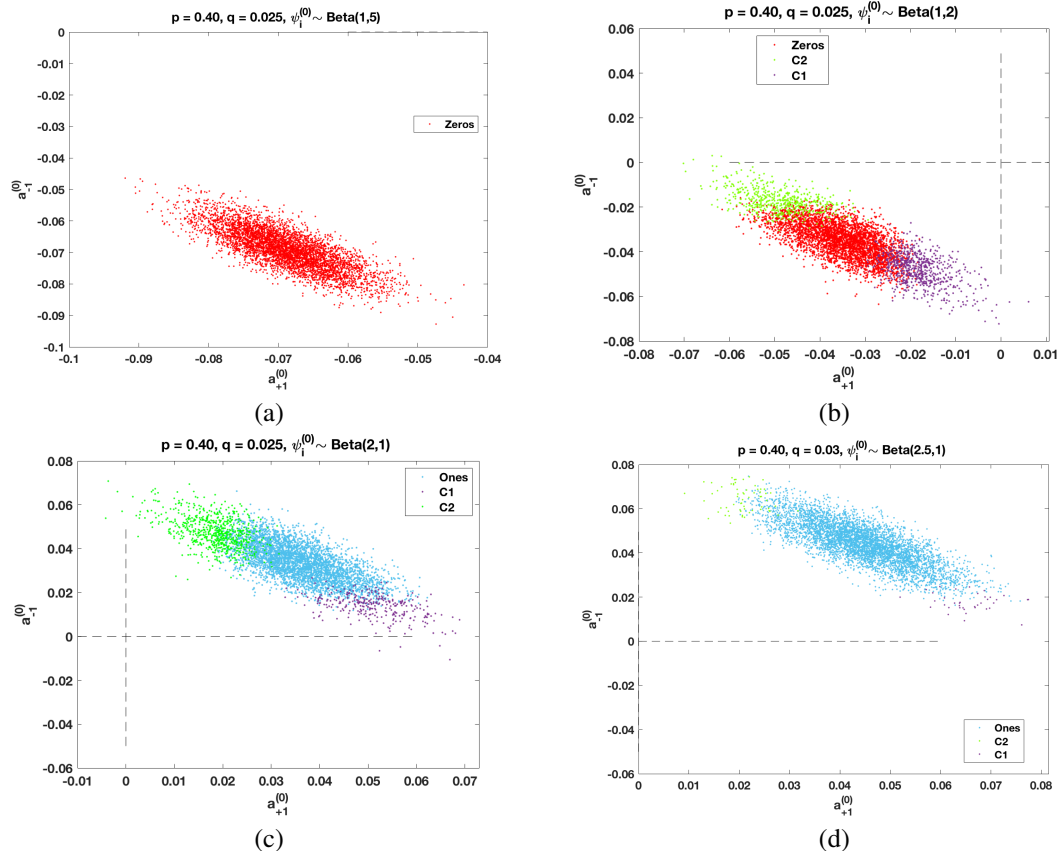

Figure 1: $n = 200$ and $5000$, $\psi^{(0)} \sim \text{Beta}(\alpha, \beta)^{\otimes n}$ for various values of $\alpha$ and $\beta$. These $\psi^{(0)}$ are mapped to $(a_{+1}^{(0)}, a_{-1}^{(0)})$ (see (5)) and plotted. $C_1$ (magenta) and $C_2$ (green) correspond to the limit points $\mathbf{1}_{C_1}$ and $\mathbf{1}_{C_2}$. Other limit points are 'Ones', i.e. $\mathbf{1}$ (blue) and 'Zeros', i.e. $\mathbf{0}$ (red).

The notable thing is, in Figure 1-(a) and (d), the Beta distribution has mean $0.16$ and $0.71$ respectively. So the initialization is more skewed towards values that are closer to zero or closer to one. In these cases most of the random runs converge to the all zeros or all ones, with very few converging to the ground truth. However, for Figure 1-(b) and (d), the mean of the Beta is $0.3$ and $0.7$, and we see considerably more convergences to the ground truth. Also, (b) and (d) are, in some sense, mirror images of each other, i.e. in one, the majority converges to $\mathbf{0}$; whereas in the other, the majority converges to $\mathbf{1}$.

In Figure 2-(a), we examine initializations of the type described in Lemma 3.3 and the resulting estimation error. For each $c_0$, we initialize $\psi^{(0)}$ such that $\mathbb{E}(\psi^{(0)}) = (1/2 + c_0)\mathbf{1}_{C_1} + (1/2 - c_0)\mathbf{1}_{C_2}$ with iid noise. The y-axis shows the average distance between $\psi^{(20)}$ and the true $Z$ from 500 such initializations, as measured by $\|\psi^{(20)} - Z\|_1/n$. For every choice of $p, q$, a network of size 400 with two equal sized blocks was generated. In all cases, sufficiently large $c_0$ guarantees convergence to the truth. We also observe that the performance deteriorates when $p - q$ becomes small, either when $p$ decreases or when the network becomes sparser.

## 5  Discussion

In this paper, we work with the BCAVI mean field variational algorithm for a simple two class stochastic blockmodel with equal sized classes. Mean field methods are used widely for their scalability. However, existing theoretical works typically analyze the behavior of the global optima, or the local convergence behavior when initialized near the ground truth. In the simple setting considered, we show two interesting results. First, we show that, when the model parameters are

known, random initializations may lead to convergence to the ground truth. In contrast, when the parameters are not known, but estimated, we show that a random initialization converges, with high probability, to a meaningless local optimum. This shows the futility of using multiple random initializations, which is typically done in practice when no prior knowledge is available.

In view of recent works on the optimization landscape for Gaussian mixtures [12, 24], we would like to comment that, despite falling into the category of latent variable models, the SBM has fundamental differences from Gaussian mixtures which require different analysis techniques. The posterior probabilities of the latent labels in the latter model can be easily estimated when the parameters are known, whereas this is not the case for SBM since the posterior probability $\mathbb{P}(Z_i|A)$ depends on the entire network. The significance of Theorem 3.3 lies in characterizing the convergence of label estimates given the correct parameters for general initializations, which is different from the type of parameter convergence shown in [12, 24]. Furthermore, as most of the existing literature for the SBM focuses on estimating the labels first, our results provide an important complementary direction by suggesting that one could start with parameter estimation instead. A natural direction is to investigate how robust the results on the known $p, q$ setting are when we can estimate $p$ and $q$ within some small error.

While we only show results for two classes, we expect that our main theoretical results generalize well to $K > 2$ and will leave the analysis for future work. As an illustration, consider a setting similar to that of Figure 1-(a) but for $n = 450$ with $K = 3$ equal sized classes. $p = 0.5$, $q = 0.01$ are known and $\psi_0$ is initialized with a Dirichlet$(0.1, 0.1, 0.1)$ distribution. Each row of the matrix in Figure 2-(b) represents a stationary cluster membership vector from a random initialization.

In Figure 2, all 1000 random initializations converge to stationary points $\psi$ lying in the span of $\{\mathbf{1}_{\mathcal{C}_1}, \mathbf{1}_{\mathcal{C}_2}, \mathbf{1}_{\mathcal{C}_3}\}$, which are the membership vectors for each class. We represent the node memberships with different colors, and there are $1 + \binom{3}{2} = 4$ different types of stationary points, not counting label permutations. Another stationary point (the all ones vector that puts everyone in the same class) can be obtained with other initialization schemes, e.g., when the rows of $\psi^{(0)}$ are identical. For a general $K$- blockmodel, we conjecture that the number of stationary points grows exponentially with $K$. Similar to Figure 1-(a), a significant fraction of the random initializations converge to the ground truth. On the other hand, when $p, q$ are unknown, random initializations always converge to the uninformative stationary point $(1/3, 1/3, 1/3)$, analogous to Lemma 3.2.

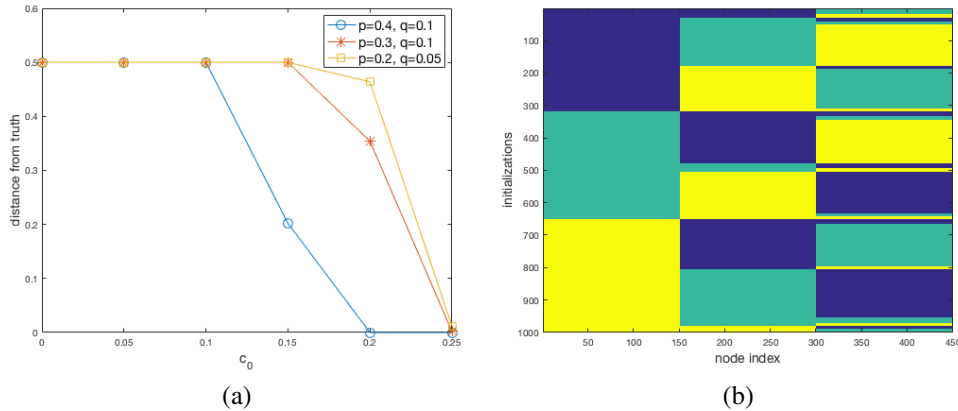

(a)                                           (b)

Figure 2: (a) Average distance between the estimated $\psi$ and the true $Z$ with respect to $c_0$, where $\mathbb{E}(\psi^{(0)}) = (1/2 + c_0)\mathbf{1}_{\mathcal{C}_1} + (1/2 - c_0)\mathbf{1}_{\mathcal{C}_2}$. (b) Convergence to stationary points for known $p, q$, $K = 3$. Rows permuted for clarity.

## Acknowledgements

SSM thanks Professor Peter J. Bickel for helpful discussions. PS is partially funded by NSF grant DMS1713082. YXRW is supported by the ARC DECRA Fellowship.

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
