[Supplementary Material]

# Appendix for "Mean Field for the Stochastic Blockmodel: Optimization Landscape and Convergence Issues"

**Soumendu Sunder Mukherjee**∗
Interdisciplinary Statistical Research Unit (ISRU)
Indian Statistical Institute, Kolkata
Kolkata 700108, India
soumendu041@gmail.com

**Purnamrita Sarkar**∗
Department of Statistics and Data Science
University of Texas, Austin
Austin, TX 78712
purna.sarkar@austin.utexas.edu

**Y. X. Rachel Wang**∗
School of Mathematics and Statistics
University of Sydney
NSW 2006, Australia
rachel.wang@sydney.edu.au

**Bowei Yan**
Department of Statistics and Data Science
University of Texas, Austin
Austin, TX 78712
boweiy@utexas.edu

## Abstract

This supplementary article contains an appendix to our paper "Mean Field for the Stochastic Blockmodel: Optimization Landscape and Convergence Issues", providing derivation of stationarity equations for the mean field log-likelihood and the proofs of our main results.

## 1   The Variational principle and mean field

We start with the following simple observation:

$$\log P(A; B, \pi) = \log \sum_Z P(A, Z; B, \pi) = \log \left( \sum_Z \frac{P(A, Z; B, \pi)}{\psi(Z)} \psi(Z) \right)$$

$$\overset{\text{(Jensen)}}{\geq} \sum_Z \log \left( \frac{P(A, Z; B, \pi)}{\psi(Z)} \right) \psi(Z) \qquad \forall \psi \text{ prob. on } \mathcal{Z}.$$

In fact, equality holds for $\psi^*(Z) = P(Z|A; B, \pi)$. Therefore, if $\Psi$ denotes the set of all probability measures on $\mathcal{Z}$, then

$$\log P(A; B, \pi) = \max_{\psi \in \Psi} \sum_Z \log \left( \frac{P(A, Z; B, \pi)}{\psi(Z)} \right) \psi(Z). \tag{A.1}$$

The crucial idea from variational inference is to replace the set $\Psi$ above by some easy-to-deal-with subclass $\Psi_0$ to get a lower bound on the log-likelihood.

$$\log P(A; B, \pi) \geq \max_{\psi \in \Psi_0 \subset \Psi} \sum_Z \log \left( \frac{P(A, Z; B, \pi)}{\psi(Z)} \right) \psi(Z). \tag{A.2}$$

Also the optimal $\psi_\star \in \Psi_0$ is a potential candidate for an estimate of $P(Z|A; B, \pi)$. Estimating $P(Z|A; B, \pi)$ is profitable since then we can obtain an estimate of the community membership

---

∗Equal contribution.

matrix by setting $Z_{ia} = 1$ for the $i$th agent where

$$a = \arg\max_b P(Z_{ib} = 1 | A; B, \pi). \tag{A.3}$$

The goal now has become optimizing the lower bound in (A.2).

## 2 Derivation of stationarity equations

$$
\begin{aligned}
\frac{\partial \ell}{\partial \psi_i} &= 4t \sum_{j:j \neq i} (\psi_j - \frac{1}{2})(A_{ij} - \lambda) - \log\left(\frac{\psi_i}{1 - \psi_i}\right), \\
\frac{\partial \ell}{\partial p} &= \frac{1}{2} \sum_{i,j:i \neq j} (\psi_i \psi_j + (1 - \psi_i)(1 - \psi_j))\left(A_{ij}\left(\frac{1}{p} + \frac{1}{1-p}\right) - \frac{1}{1-p}\right), \\
\frac{\partial \ell}{\partial q} &= \frac{1}{2} \sum_{i,j:i \neq j} (\psi_i(1 - \psi_j) + (1 - \psi_i)\psi_j)\left(A_{ij}\left(\frac{1}{q} + \frac{1}{1-q}\right) - \frac{1}{1-q}\right).
\end{aligned} \tag{A.4}
$$

Therefore

$$
\begin{aligned}
\frac{\partial^2 \ell}{\partial \psi_j \partial \psi_i} &= 4t(A_{ij} - \lambda)(1 - \delta_{ij}) - \frac{1}{\psi_i(1 - \psi_i)}\delta_{ij}, \\
\frac{\partial^2 \ell}{\partial \psi_i \partial p} &= \frac{1}{2} \sum_{j:j \neq i} \left(\frac{1}{2} - \psi_j\right)\left(A_{ij}\left(\frac{1}{p} + \frac{1}{1-p}\right) - \frac{1}{1-p}\right), \\
\frac{\partial^2 \ell}{\partial \psi_i \partial q} &= \frac{1}{2} \sum_{j:j \neq i} \left(\psi_i - \frac{1}{2}\right)\left(A_{ij}\left(\frac{1}{q} + \frac{1}{1-q}\right) - \frac{1}{1-q}\right), \\
\frac{\partial^2 \ell}{\partial p^2} &= \frac{1}{2} \sum_{i,j:i \neq j} (\psi_i \psi_j + (1 - \psi_i)(1 - \psi_j))\left(A_{ij}\left(-\frac{1}{p^2} + \frac{1}{(1-p)^2}\right) - \frac{1}{(1-p)^2}\right), \\
\frac{\partial^2 \ell}{\partial q^2} &= \frac{1}{2} \sum_{i,j:i \neq j} (\psi_i(1 - \psi_j) + (1 - \psi_i)\psi_j)\left(A_{ij}\left(-\frac{1}{q^2} + \frac{1}{(1-q)^2}\right) - \frac{1}{(1-q)^2}\right), \\
\frac{\partial^2 \ell}{\partial q \partial p} &= 0.
\end{aligned} \tag{A.5}
$$

## 3 Proofs of main results

*Proof of Proposition 3.1.* For any $a > b > 0$, we have

$$\frac{a - b}{a} < \log\left(\frac{a}{b}\right) < \frac{a - b}{b},$$

which can be proved using the inequality $\log(1 + x) < x$ for $x > -1, x \neq 0$. Therefore

$$\frac{p - q}{p} < \log\left(\frac{p}{q}\right) < \frac{p - q}{q}, \quad \text{and} \quad \frac{p - q}{1 - q} < \log\left(\frac{1 - q}{1 - p}\right) < \frac{p - q}{1 - p}.$$

So

$$\frac{(p - q)(1 + p - q)}{2(1 - q)p} < t = \frac{1}{2}\left(\log\left(\frac{p}{q}\right) + \log\left(\frac{1 - q}{1 - p}\right)\right) < \frac{(p - q)(1 - p + q)}{2(1 - p)q},$$

and

$$q = \frac{\frac{p-q}{1-q}}{\frac{p-q}{q} + \frac{p-q}{1-q}} < \lambda = \frac{\log(\frac{1-q}{1-p})}{\log(\frac{p}{q}) + \log(\frac{1-q}{1-p})} < \frac{\frac{p-q}{1-p}}{\frac{p-q}{p} + \frac{p-q}{1-p}} = p.$$

This completes the proof. □

## 3.1 Proofs of results in Section 3.1

*Proof of Proposition 3.2.* That $\psi = \frac{1}{2}\mathbf{1}$ is a stationary point is obvious from the stationarity equations (A.4). The eigenvalues of $-4I + 4tM$, the Hessian at $\frac{1}{2}\mathbf{1}$, are $h_i = -4 + 4t\nu_i$. We have $\nu_1 = n\alpha_+ - (p - \lambda) = \Theta(n)$, and hence so is $h_1$. Also, $p - \lambda > 0$, so that $\nu_3 < 0$, and hence $h_3 < 0$. Thus we have two eigenvalues of the opposite sign. $\qquad\square$

*Proof of Theorem 3.3.* From (5), we have

$$\psi_i^{(s+1)} = g(na_{\sigma_i}^{(s)} + b_i^{(s)}) = g(na_{\sigma_i}^{(s)}) + \delta_i^{(s)},$$

where $|\delta_i^{(s)}| = O(\exp(-n|a_{\sigma_i}^{(s)}|))$, where we have used the fact that

$$g(nx + y) - g(nx) = g(nx)g(nx + y)(e^y - 1)\exp(-(nx + y)).$$

Writing as a vector, we have

$$\psi^{(s+1)} = g(na_{+1}^{(s)})\mathbf{1}_{\mathcal{C}_1} + g(na_{-1}^{(s)})\mathbf{1}_{\mathcal{C}_2} + \delta^{(s)}, \qquad (A.6)$$

where $\|\delta^{(s)}\|_\infty = \max_i |\delta_i^{(s)}| = O(\exp(-n\min\{|a_{+1}^{(s)}|, |a_{-1}^{(s)}|\}))$. Note that by our assumption, $\|\delta^{(0)}\|_\infty = O(\exp(-n\min\{|a_{+1}^{(s)}|, |a_{-1}^{(s)}|\})) = o(1)$. Now

$$\zeta_1^{(s+1)} = \frac{\langle \psi^{(s+1)}, u_1 \rangle}{n} = \frac{g(na_{+1}^{(s)}) + g(na_{-1}^{(s)})}{2} + O(\|\delta^{(s)}\|_\infty),$$

and

$$\zeta_2^{(s+1)} = \frac{\langle \psi^{(s+1)}, u_2 \rangle}{n} = \frac{g(na_{+1}^{(s)}) - g(na_{-1}^{(s)})}{2} + O(\|\delta^{(s)}\|_\infty).$$

Note that $g(na_{\pm 1}^{(s)}) = \mathbf{1}_{\{a_{\pm 1}^{(s)} > 0\}} + O(\|\delta^{(s)}\|_\infty)$. Now, using (A.6), we have

$$
\begin{aligned}
&\frac{\|\psi^{(s+1)} - \ell(\psi^{(0)})\|_2^2}{n} \\
&= \frac{\|(g(na_{+1}^{(s)}) - \mathbf{1}_{\{a_{+1}^{(0)} > 0\}})\mathbf{1}_{\mathcal{C}_1} + (g(na_{-1}^{(s)}) - \mathbf{1}_{\{a_{-1}^{(0)} > 0\}})\mathbf{1}_{\mathcal{C}_2} + \delta^{(s)}\|^2}{n} \\
&\leq \frac{2(\|(g(na_{+1}^{(s)}) - \mathbf{1}_{\{a_{+1}^{(0)} > 0\}})\mathbf{1}_{\mathcal{C}_1}\|_2^2 + \|(g(na_{-1}^{(s)}) - \mathbf{1}_{\{a_{-1}^{(0)} > 0\}})\mathbf{1}_{\mathcal{C}_2}\|_2^2 + \|\delta^{(s)}\|^2)}{n} \\
&\leq |g(na_{+1}^{(s)}) - \mathbf{1}_{\{a_{+1}^{(0)} > 0\}}|^2 + |g(na_{-1}^{(s)}) - \mathbf{1}_{\{a_{-1}^{(0)} > 0\}}|^2 + 2\|\delta^{(s)}\|_\infty^2 \\
&= |\mathbf{1}_{\{a_{+1}^{(s)} > 0\}} - \mathbf{1}_{\{a_{+1}^{(0)} > 0\}}|^2 + |\mathbf{1}_{\{a_{+1}^{(s)} > 0\}} - \mathbf{1}_{\{a_{-1}^{(0)} > 0\}}|^2 + O(\|\delta^{(s)}\|_\infty^2). \qquad (A.7)
\end{aligned}
$$

From the above representation and our assumption on $n|a_{\pm 1}^{(0)}|$, the bound for $s = 1$ follows. We will now consider the four different cases of different signs of $a_{\pm 1}^{(s)}$.

**Case 1:** $a_1^{(s)} > 0, a_{-1}^{(s)} > 0$. In this case $g(na_1^{(s)}) = g(na_{-1}^{(s)}) = 1 + O(\|\delta^{(s)}\|_\infty)$, so that

$$(\zeta_1^{(s+1)}, \zeta_2^{(s+1)}) = (1, 0) + O(\|\delta^{(s)}\|_\infty).$$

This implies that

$$a_{\pm 1}^{(s+1)} = 2t\alpha_+ + O(\|\delta^{(s)}\|_\infty).$$

If $\alpha_+ > 0$, $a_{\pm 1}^{(s+1)}$ have the same sign as $a_{\pm 1}^{(s)}$. Otherwise, if $\alpha_+ < 0$, both of them become negative (and we thus have to go to Case 2 below). Note that, here and in the subsequent cases, we are using that fact that $\|\delta^{(s)}\|_\infty = o(1)$, for $s = 0$, by our assumption and it stays the same for $s \geq 1$ because of relations like the above (that is $a_{\pm 1}^{(1)} = -2t\alpha_+ + o(1)$, so that $\|\delta^{(1)}\|_\infty = \exp(-n\min\{|a_{+1}^{(1)}|, |a_{-1}^{(1)}|\}) = O(\exp(-Cnt\alpha_+)) = o(1)$, and so on).

**Case 2:** $a_1^{(s)} < 0, a_{-1}^{(s)} < 0$. In this case $1 - g(na_1^{(s)}) = 1 - g(na_{-1}^{(s)}) = 1 + O(\|\delta^{(s)}\|_\infty)$, so that

$$(\zeta_1^{(s+1)}, \zeta_2^{(s+1)}) = (0, 0) + O(\|\delta^{(s)}\|_\infty).$$

This implies that

$$a_{\pm 1}^{(s+1)} = -2t\alpha_+ + O(\|\delta^{(s)}\|_\infty).$$

If $\alpha_+ > 0$, $a_{\pm 1}^{(s+1)}$ have the same sign as $a_{\pm 1}^{(s)}$. Otherwise, if $\alpha_+ < 0$, both of them become positive (and we thus have to go to Case 1 above).

**Case 3:** $a_1^{(s)} > 0, a_{-1}^{(s)} < 0$. In this case $g(na_1^{(s)}) = 1 - g(na_{-1}^{(s)}) = 1 + O(\|\delta^{(s)}\|_\infty)$, so that

$$(\zeta_1^{(s+1)}, \zeta_2^{(s+1)}) = (\frac{1}{2}, \frac{1}{2}) + O(\|\delta^{(s)}\|_\infty).$$

This implies that

$$a_{\pm 1}^{(s+1)} = \pm 2t\alpha_- + O(\|\delta^{(s)}\|_\infty).$$

Since $\alpha_- > 0$, $a_{\pm 1}^{(s+1)}$ have the same sign as $a_{\pm 1}^{(s)}$.

**Case 4:** $a_1^{(s)} < 0, a_{-1}^{(s)} > 0$. In this case $1 - g(na_1^{(s)}) = g(na_{-1}^{(s)}) = 1 + O(\|\delta^{(s)}\|_\infty)$, so that

$$(\zeta_1^{(s+1)}, \zeta_2^{(s+1)}) = (\frac{1}{2}, -\frac{1}{2}) + O(\|\delta^{(s)}\|_\infty).$$

This implies that

$$a_{\pm 1}^{(s+1)} = \mp 2t\alpha_- + O(\|\delta^{(s)}\|_\infty).$$

Since $\alpha_- > 0$, $a_{\pm 1}^{(s+1)}$ have the same sign as $a_{\pm 1}^{(s)}$.

Note that, in the case $\alpha_+ = 0$, $a_{\pm 1}^{(s)} = \pm 4t\zeta_2^{(s)}\alpha_-$, so that $a_{\pm 1}^{(s)}$ have opposite signs and we land in Cases 3 or 4.

We conclude that, if $\alpha_+ \geq 0$, then we stay in the same case where we began, and otherwise if $\alpha_+ < 0$ we have a cycling behavior between Cases 1 and 2. Now the desired conclusion follows from the bound (A.7).

In the proof above, we can allow sparser graphs, with $p, q \gg \frac{1}{n}$. More explicitly, let $p = \rho_n a, q = \rho_n b$, with $a > b > 0$ and $\rho_n \gg \frac{1}{n}$. Then, $t = \Omega(1)$, and $\alpha_+ \leq p - q = \rho_n(a-b), \alpha_- = (p-q)/2 = \rho_n(a-b)/2$. So, we do have $nt|\alpha_\pm| \to \infty$. □

*Proof of Theorem 3.4.* We begin by noting that $\widehat{M} - M = A - \mathbb{E}(A|Z) := A - \tilde{P}$. For the first iteration, we rewrite the sample iterations (7) as

$$\hat{\xi}^{(1)} = 4tM\left(\psi^{(0)} - \frac{1}{2}\mathbf{1}\right) + 4t(\widehat{M} - M)\left(\psi^{(0)} - \frac{1}{2}\mathbf{1}\right)$$

$$= \xi^{(1)} + \underbrace{4t(A - \tilde{P})\left(\psi^{(0)} - \frac{1}{2}\mathbf{1}\right)}_{=:nr^{(0)}}.$$

Therefore, similar to the population case, we have

$$\hat{\psi}_i^{(1)} = g(na_{\sigma_i}^{(0)} + b_i^{(0)} + nr_i^{(0)}).$$

Note that

$$r_i^{(0)} = \frac{4t}{n}\sum_{j\neq i}(A_{ij} - \tilde{P}_{ij}|Z_i, Z_j)(\psi_j^{(0)} - \frac{1}{2}). \tag{A.8}$$

Assume that $\psi^{(0)}$ is independent of $A$. Since our probability statements will be with respect to the randomness in $A$, we may assume that $\psi^{(0)}$ is fixed. Let $Y_{ij} = (A_{ij} - \tilde{P}_{ij})(\psi_j^{(0)} - \frac{1}{2})$. Then

the $Y_{ij}$ are independent random variables for $j \neq i$, and $\mathbb{E}(Y_{ij}) = 0$. Also, $|Y_{ij}| \leq |\psi_j^{(0)} - \frac{1}{2}| \leq \|\psi^{(0)} - \frac{1}{2}\|_\infty = \Delta$, say, and $\mathbb{E}Y_{ij}^2 = (\psi_j^{(0)} - \frac{1}{2})^2 \text{Var}(A_{ij}) = O(\rho_n(\psi_j^{(0)} - \frac{1}{2})^2)$. So, by Bernstein's inequality,

$$\mathbb{P}(\frac{1}{n}\sum_{j \neq i} Y_{ij} > \epsilon) \leq \exp\left(\frac{-\frac{1}{2}n^2\epsilon^2}{\sum_{j \neq i}\mathbb{E}Y_{ij}^2 + \frac{1}{3}\Delta n\epsilon}\right)$$

$$\leq \exp\left(\frac{-\frac{1}{2}n^2\epsilon^2}{C\rho_n\|\psi^{(0)} - \frac{1}{2}\|_2^2 + \frac{1}{3}\Delta n\epsilon}\right)$$

$$\leq \exp\left(\frac{-\frac{1}{2}n^2\epsilon^2}{Cn\rho_n\Delta^2 + \frac{1}{3}\Delta n\epsilon}\right). \tag{A.9}$$

It follows from here that $nr_i^{(0)} = O(\sqrt{n\rho_n}\Delta\log n)$ with high probability, if $\sqrt{n\rho_n} = \Omega(\log n)$. In fact, by taking a suitably large constant in the big "Oh", we can show, via a union bound, that $\max_i nr_i^{(0)} = O(\sqrt{n\rho_n}\Delta\log n)$ with high probability.

Now, from our assumption $n|a_{\pm 1}^{(0)}| \gg \max\{\sqrt{n\rho_n}\|\psi^{(0)} - \frac{1}{2}\|_\infty\log n, 1\}$, it follows that $na_{\sigma_i}^{(0)} \gg nr_i^{(0)} + b_i^{(0)}$ with high probability, simultaneously for all $i$. Thus, similar to the population case, we can write

$$\hat{\psi}^{(1)} = g(na_{+1}^{(0)})\mathbf{1}_{\mathcal{C}_1} + g(na_{-1}^{(0)})\mathbf{1}_{\mathcal{C}_2} + \hat{\delta}^{(0)},$$

where $\|\hat{\delta}^{(0)}\|_\infty = O(\exp(-n\min\{|a_{+1}^{(0)}|, |a_{-1}^{(0)}|\})) = o(1)$, with high probability. After this the proof proceeds like the the proof of Theorem 3.3, and so we omit it.

Let us consider the case with $s = 2$ and we will show $nr_i^{(1)}$ can be bounded in a general way. Now

$$\xi^{(2)} = 4t(A - \lambda(J - I))(\hat{\psi}^{(1)} - 1/2)$$

$$= 4tM(\hat{\psi}^{(1)} - 1/2) + nr^{(1)}$$

$$= 4tM(\hat{\psi}^{(1)} - 1/2) + \underbrace{4t(A - \tilde{P})(\hat{\psi}^{(1)} - \ell(\psi^{(0)}))}_{R_1} + \underbrace{4t(A - \tilde{P})(\ell(\psi^{(0)}) - \frac{1}{2}\mathbf{1})}_{R_2}.$$

Now the analysis of the first term follows from Theorem 3.3. It is also easy to see $\max_i |R_{2,i}| = O_P(\sqrt{n\rho_n})$, since $\ell(\psi^{(0)}) \in \{\mathbf{1}_{\mathcal{C}_1}, \mathbf{1}_{\mathcal{C}_2}, \mathbf{1}, \mathbf{0}, \frac{1}{2}\mathbf{1}\}$. For $R_1$,

$$\max_i |R_{1,i}| \leq \|R_1\|_2 \leq \|A - \tilde{P}\|_{op}\|\hat{\psi}^{(1)} - \tilde{\ell}(\psi^{(0)})\|_2$$

$$= O_P(\sqrt{n\rho_n})\sqrt{n} \cdot O(\exp(-\Theta(n\min\{|a_{+1}^{(0)}|, |a_{-1}^{(0)}|\}))) = o_P(1),$$

under our assumption that $n|a_{\pm 1}^{(0)}| \gg \max\{\sqrt{n\rho_n}\|\psi^{(0)} - \frac{1}{2}\|_\infty\log n, 1\}$. Hence $\max_i |nr_i^{(1)}| = O_P(\sqrt{n\rho_n})$, and $na_{\sigma_i}^{(1)} \gg nr_i^{(1)} + b_i^{(1)}$ with high probability, simultaneously for all $i$. The same analysis as in the $s = 1$ case follows.

The case for general $s$ can be proved by induction using the same decomposition of $nr^{(s)}$, replacing $\ell(\psi^{(0)})$ with a more general $\tilde{\ell}(\psi^{(0)}) \in \{\ell(\psi^{(0)}), \mathbf{0}, \mathbf{1}\}$ depending on the signs of $a_{+1}^{(0)}, a_{-1}^{(0)}, \alpha_+$ as described in Theorem 3.3 for $s \geq 2$. $\square$

*Proof of Corollary 3.5.* From Theorem 3.3, it follows that, when $\alpha_+ > 0$,

$$\mathfrak{M}(\mathcal{S}_1) \geq \mathfrak{M}(\{\psi^{(0)} \mid a_{+1}^{(0)} > 0, a_{-1}^{(0)} > 0, na_{\pm 1}^{(0)} \gg 1\}$$

$$= \mathfrak{M}(\{\psi^{(0)} \mid a_{+1}^{(0)} \gg \frac{1}{n}, a_{-1}^{(0)} \gg \frac{1}{n}\})$$

$$\geq \mathfrak{M}(\{\psi^{(0)} \mid a_{+1}^{(0)} > \frac{1}{n^\gamma}, a_{-1}^{(0)} > \frac{1}{n^\gamma}\}),$$

for any $0 < \gamma < 1$ and so on for the other other limit points.

More explicitly,

$$\{\psi^{(0)} \mid a_{+1}^{(0)} > \frac{1}{n^\gamma}, a_{-1}^{(0)} > \frac{1}{n^\gamma}\} = \{\psi^{(0)} \mid (\zeta_1^{(0)} - \frac{1}{2})\alpha_+ + \zeta_2^{(0)}\alpha_- > \frac{1}{4tn^\gamma},$$

$$(\zeta_1^{(0)} - \frac{1}{2})\alpha_+ - \zeta_2^{(0)}\alpha_- > \frac{1}{4tn^\gamma}\}$$

$$= H_+^\gamma \cap H_-^\gamma \cap [0,1]^n,$$

All in all, we have

$$\mathfrak{M}(\mathcal{S}_1) \geq \lim_{\gamma \uparrow 1} \mathfrak{M}(H_+^\gamma \cap H_-^\gamma \cap [0,1]^n).$$

This completes the proof. $\qquad\square$

## 3.2 Proofs of results in Section 3.2

*Proof of Proposition 3.6.* That the described point is a stationary point is easy to verify, because of the presence of the $(\psi_i - \frac{1}{2})$ terms in the stationarity equations (A.4). Now, from (A.5), we see that the Hessian matrix at $(\frac{1}{2}\mathbf{1}, \frac{\mathbf{1}^\top A\mathbf{1}}{n(n-1)}, \frac{\mathbf{1}^\top A\mathbf{1}}{n(n-1)}, \frac{1}{2})$ is given by

$$H = \begin{pmatrix} -4I & \mathbf{0} & \mathbf{0} \\ \mathbf{0}^\top & -\frac{n(n-1)}{4\hat{a}(1-\hat{a})} & 0 \\ \mathbf{0}^\top & 0 & -\frac{n(n-1)}{4\hat{a}(1-\hat{a})} \end{pmatrix},$$

where $\hat{a} = \frac{\mathbf{1}^\top A\mathbf{1}}{n(n-1)}$. Clearly, $H$ is negative definite. This completes the proof. $\qquad\square$

*Proof of Lemma 3.1.* First note that conditioning on the true labels $Z$, $\mathbb{E}(A|Z) = \tilde{P}$. For the update of $p^{(1)}$, we have

$$p^{(1)} = \frac{\psi^T \tilde{P}\psi + (\mathbf{1} - \psi)^T \tilde{P}(\mathbf{1} - \psi)}{\psi^T(J - I)\psi + (\mathbf{1} - \psi)^T(J - I)(\mathbf{1} - \psi)}$$
$$+ \frac{\psi^T(A - \tilde{P})\psi + (\mathbf{1} - \psi)^T(A - \tilde{P})(\mathbf{1} - \psi)}{\psi^T(J - I)\psi + (\mathbf{1} - \psi)^T(J - I)(\mathbf{1} - \psi)},$$

where the first term can be written as

$$\frac{\psi^T(\frac{p+q}{2}u_1 u_1^T + \frac{p-q}{2}u_2 u_2^T - pI)\psi + (\mathbf{1} - \psi)^T(\frac{p+q}{2}u_1 u_1^T + \frac{p-q}{2}u_2 u_2^T - pI)(\mathbf{1} - \psi)}{\psi^T(u_1 u_1^T - I)\psi + (\mathbf{1} - \psi)^T(u_1 u_1^T - I)(\mathbf{1} - \psi)}$$
$$= \frac{\frac{p+q}{2}n^2(\zeta_1^2 + (1 - \zeta_1)^2) + n^2(p - q)\zeta_2^2 - px}{\zeta_1^2 n^2 + (1 - \zeta_1)^2 n^2 - x}$$
$$= \frac{p+q}{2} + \frac{(p-q)(\zeta_2^2 - x/2n^2)}{\zeta_1^2 + (1 - \zeta_1)^2 - x/n^2},$$

where $x = \psi^T\psi + (\mathbf{1} - \psi)^T(\mathbf{1} - \psi) \geq n^2/4$. The second term can be bounded by noting $\mathbb{E}(\psi^T(A - \tilde{P})\psi) = 0$ and $\text{Var}(\psi^T(A - \tilde{P})\psi) \leq 2n(n-1)p$. By Chebyshev's inequality, $\psi^T(A - \tilde{P})\psi = O_P(\sqrt{\rho_n}n)$.

This is because

$$\mathbb{E}_{\psi,A}[\psi^T(A - \tilde{P})\psi] = \mathbb{E}_\psi \mathbb{E}_A[\psi^T(A - \tilde{P})\psi \mid \psi] = 0,$$

and

$$\text{Var}_{\psi,A}[\psi^T(A - \tilde{P})\psi] = \mathbb{E}\text{Var}(\psi^T(A - \tilde{P})\psi \mid \psi) + \text{Var}(\mathbb{E}[\psi^T(A - \tilde{P})\psi \mid \psi])$$
$$= \mathbb{E}\text{Var}(\psi^T(A - \tilde{P})\psi \mid \psi)$$
$$= 4\mathbb{E}\sum_{i<j} \psi_i \psi_j \text{Var}(A_{ij}) \leq 2n(n-1)p.$$

$(1-\psi)^T(A - \tilde{P})(1-\psi)$ can be handled similarly, and

$$\psi^T(J-I)\psi + (\mathbf{1}-\psi)^T(J-I)(\mathbf{1}-\psi)$$

$$= \left(\sum_i \psi_i\right)^2 + \left(n - \sum_i \psi_i\right)^2 - \psi^T\psi - (\mathbf{1}-\psi)^T(\mathbf{1}-\psi)$$

$$\geq n^2/2 - 2n,$$

since the first two terms are minimized at $\sum_i \psi_i = n/2$.

The result for $q^{(1)}$ is proved analogously. $\qquad\square$

*Proof of Proposition 3.7.* Let $\psi = \zeta_1 u_1 + \zeta_2 u_2 + w$, $w \in \text{span}\{u_1, u_2\}^\perp$, be a stationary point. We will consider the population version of all the updates and replace $A$ with $\mathbb{E}(A|Z) := \tilde{P}$ and $\rho_n \to 0$. By Lemma 3.1,

$$\tilde{p} = \frac{p+q}{2} + \underbrace{\frac{(p-q)(\zeta_2^2 - x/2n^2)}{\zeta_1^2 + (1-\zeta_1)^2 - x/n^2}}_{\epsilon_1'},$$

$$\tilde{q} = \frac{p+q}{2} - \underbrace{\frac{(p-q)(\zeta_2^2 + y/2n^2)}{2\zeta_1(1-\zeta_1) - y/n^2}}_{\epsilon_2'}. \qquad (A.10)$$

In this case, the update equation (4) becomes

$$\xi = 4\tilde{t}(\tilde{P} - \tilde{\lambda}(J-I))(\psi^{(s)} - \frac{1}{2}\mathbf{1})$$

$$= 4\tilde{t}n\left(\left(\zeta_1 - \frac{1}{2}\right)\left(\frac{p+q}{2} - \tilde{\lambda}\right)u_1 + \frac{p-q}{2}\zeta_2 u_2\right) + 4\tilde{t}(\tilde{\lambda} - p)\left(\psi - \frac{1}{2}\mathbf{1}\right)$$

$$:= n\tilde{a} + \tilde{b} \qquad (A.11)$$

where $\tilde{\lambda}$ and $\tilde{t}$ are defined in terms of $\tilde{p}$ and $\tilde{q}$. Since $\psi$ is a stationary point, the above update gives $\psi = g(\xi)$.

We consider the following cases.

**Case 1:** $\zeta_2^2 = \Omega(1)$. Since $\zeta_1(1-\zeta_1) \geq \zeta_2^2$, it is easy to see that (A.10) implies that $\tilde{p} > \frac{p+q}{2} > \tilde{q}$, thus $\tilde{p} - \tilde{q} = \Omega(\rho_n)$, $\tilde{t} = \Omega(1)$, $\tilde{p} < \tilde{\lambda} < \tilde{q}$. It follows then $\tilde{b}_i = O(\rho_n)$, and $|\tilde{a}_i| = \Omega(\rho_n)$ for $i \in \mathcal{C}_1$ or $i \in \mathcal{C}_2$ (or both). In any of these cases, $\|w\| = O(\rho_n\sqrt{n}) = o(\sqrt{n})$.

**Case 2:** $\zeta_2 = o(1)$. Note that $\psi^T(\mathbf{1}-\psi) \geq 0$ implies that $\zeta_1(1-\zeta_1) - \frac{\|w\|^2}{n} \geq \zeta_2^2$. If $\|w\|^2 = o(n)$, we are done. If $\|w\|^2 = \Omega(n)$, $\zeta_1(1-\zeta_1) = \Omega(1)$. In this case, $\tilde{p} = \frac{p+q}{2} + O(\rho_n\zeta_2^2)$, and similarly for $\tilde{q}$. It follows then that $\tilde{t} = O(\zeta_2^2) = o(1)$, $\tilde{\lambda} = \frac{p+q}{2} + o(\rho_n)$ (we defer the details to (A.14)- (A.18)). Also note that $\tilde{b}_i = O(\rho_n\zeta_2^2)$. When $n|\tilde{a}_i| \gg \tilde{b}_i$, $g(\xi_i) = g(n\tilde{a}_i) + o(1)$. Since $g(n\tilde{a}) \in \text{span}\{u_1, u_2\}$, this implies that $\|w\| = o(\sqrt{n})$. When $n|\tilde{a}_i| \asymp \tilde{b}_i$, $\xi_i = o(1)$, and so we have $\|w\| = o(\sqrt{n})$ again. $\qquad\square$

*Proof of Lemma 3.2.* Let $a = (p+q)/2$. By (5), define $\kappa_1 := 4t\left(\zeta_1 - \frac{1}{2}\right)(a - \lambda)$ and $\kappa_2 = 4t\zeta_2\frac{p-q}{2}$. Consider the initial distribution $\psi^{(0)}(i) \overset{iid}{\sim} f_\mu$, where $f$ is a distribution supported on $(0,1)$ with mean $\mu$. Note that we have the following:

$$\zeta_1 = \frac{\psi^T \mathbf{1}}{n} = \mu + O_P(1/\sqrt{n}), \qquad (A.12)$$

$$\zeta_2 = \frac{\psi^T u_2}{n} = O_P(1/\sqrt{n}).$$

Now using (10), recall that

$$p^{(1)} = \frac{p+q}{2} + \underbrace{\underbrace{\frac{(p-q)(\zeta_2^2 - x/2n^2)}{\zeta_1^2 + (1-\zeta_1)^2 - x/n^2}}_{\epsilon_1'} + O_P(\sqrt{\rho_n}/n)}_{\epsilon_1},$$

$$q^{(1)} = \frac{p+q}{2} - \underbrace{\underbrace{\frac{(p-q)(\zeta_2^2 + y/2n^2)}{2\zeta_1(1-\zeta_1) - y/n^2}}_{\epsilon_2'} - O_P(\sqrt{\rho_n}/n)}_{\epsilon_2}. \tag{A.13}$$

This gives

$$\epsilon_1 = \epsilon_1' + O_P\left(\frac{\sqrt{\rho_n}}{n}\right) = O_P\left(\frac{\rho_n}{n}\right) + O_P\left(\frac{\sqrt{\rho_n}}{n}\right) = O_P\left(\frac{\sqrt{\rho_n}}{n}\right),$$

$$\epsilon_2 = \epsilon_2' + O_P\left(\frac{\sqrt{\rho_n}}{n}\right) = O_P\left(\frac{\sqrt{\rho_n}}{n}\right).$$

We will use the following logarithmic inequalities for $a > \epsilon > 0$:

$$\frac{2\epsilon}{a+\epsilon} \le \log\frac{a+\epsilon}{a-\epsilon} \le \frac{2\epsilon}{a-\epsilon}. \tag{A.14}$$

Now we have

$$t = \frac{1}{2}\left(\log\left(\frac{a+\epsilon_1}{a-\epsilon_2}\right) + \log\left(\frac{1-a+\epsilon_2}{1-a-\epsilon_1}\right)\right),$$

$$2t \ge \frac{\epsilon_1+\epsilon_2}{a+\epsilon_1} + \frac{\epsilon_1+\epsilon_2}{1-a+\epsilon_2} \ge \frac{(\epsilon_1+\epsilon_2)}{(a+\epsilon_1)(1-a+\epsilon_2)},$$

$$2t \le \frac{(\epsilon_1+\epsilon_2)}{(a-\epsilon_2)(1-a-\epsilon_1)}. \tag{A.15}$$

For $\lambda$, if $\epsilon_1 + \epsilon_2 \ge 0$, we have

$$\lambda = \frac{\log\frac{1-q^{(1)}}{1-p^{(1)}}}{\log\frac{p^{(1)}}{q^{(1)}} + \log\frac{1-q^{(1)}}{1-p^{(1)}}} \le \frac{\epsilon_1+\epsilon_2}{1-a-\epsilon_1}\bigg/\left(\frac{\epsilon_1+\epsilon_2}{a+\epsilon_1} + \frac{\epsilon_1+\epsilon_2}{1-a-\epsilon_1}\right) = a+\epsilon_1. \tag{A.16}$$

$$\lambda \ge \frac{\epsilon_1+\epsilon_2}{1-a+\epsilon_2}\bigg/\left(\frac{\epsilon_1+\epsilon_2}{a-\epsilon_2} + \frac{\epsilon_1+\epsilon_2}{1-a+\epsilon_2}\right) = a-\epsilon_2. \tag{A.17}$$

If $\epsilon_1 + \epsilon_2 \le 0$,

$$\lambda = \frac{\log\frac{1-q^{(1)}}{1-p^{(1)}}}{\log\frac{p^{(1)}}{q^{(1)}} + \log\frac{1-q^{(1)}}{1-p^{(1)}}} \ge \frac{\epsilon_1+\epsilon_2}{1-a-\epsilon_1}\bigg/\left(\frac{\epsilon_1+\epsilon_2}{a+\epsilon_1} + \frac{\epsilon_1+\epsilon_2}{1-a-\epsilon_1}\right) = a+\epsilon_1, \tag{A.18}$$

$$\lambda \le \frac{\epsilon_1+\epsilon_2}{1-a+\epsilon_2}\bigg/\left(\frac{\epsilon_1+\epsilon_2}{a-\epsilon_2} + \frac{\epsilon_1+\epsilon_2}{1-a+\epsilon_2}\right) = a-\epsilon_2.$$

Now we are ready to estimate $\xi_i$. We define:

$$\kappa_1 = 4t\left(\zeta_1 - \frac{1}{2}\right)(a-\lambda) \le \left|\frac{2(\epsilon_1+\epsilon_2)}{(a-\epsilon_2)(1-a-\epsilon_1)}\left(\mu - \frac{1}{2} + O_P(1/\sqrt{n})\right)\max(|\epsilon_1|,|\epsilon_2|)\right|$$

$$\le \frac{4\max\{\epsilon_1^2,\epsilon_2^2\}}{a(1-a) + O_P(\sqrt{\rho_n}/n)}\left|\mu - \frac{1}{2} + O_P(1/\sqrt{n})\right| = O_P(1/n^2),$$

$$\kappa_2 = 4t\zeta_2\frac{(p-q)}{2} \le \left|\frac{2(\epsilon_1+\epsilon_2)}{(a-\epsilon_2)(1-a-\epsilon_1)}(p-q)O_P\left(\frac{1}{\sqrt{n}}\right)\right|$$

$$\le \frac{4\max(|\epsilon_1|,|\epsilon_2|)}{a(1-a) + O_P(\sqrt{\rho_n}/n)}(p-q)O_P(1/\sqrt{n}) = O_P(\sqrt{\rho_n}/n^{3/2}). \tag{A.19}$$

From (5) and adding the noise term from the sample version of the update,

$$\xi_i^{(1)} = n(\kappa_1 + \sigma_i \kappa_2) + b_i^{(0)} + n r_i^{(0)}, \tag{A.20}$$

where $\max_i |b_i^{(0)}| = t \cdot O_P(\rho_n) = O_P(\sqrt{\rho_n/n})$, since $t = O_P(1/(n\sqrt{\rho_n}))$ by (A.15), and $\max_i |n r_i^{(0)}| = 4t \cdot O_P(\sqrt{n\rho_n} \log n) = O_P(\log n/\sqrt{n})$ if $n\rho_n \gg (\log n)^2$, following the bound in Eq (A.9)). Now applying the update for $\psi$, we have $\psi_i^{(1)} = g(\xi^{(1)}) = \frac{1}{2} + O_P(\log n/\sqrt{n})$ uniformly for all $i$. $\qquad\square$

*Proof of Lemma 3.3.* In this setting, we write $p^{(1)}, q^{(1)}$ as follows:

$$p^{(1)} = p - (p-q)\frac{\frac{\zeta_1^2 + (1-\zeta_1)^2}{2} - \zeta_2^2}{\zeta_1^2 + (1-\zeta_1)^2 - x/n^2} + O_P(\sqrt{\rho_n}/n),$$

$$q^{(1)} = q + (p-q)\frac{\zeta_1(1-\zeta_1) - \zeta_2^2 - y/n^2}{2\zeta_1(1-\zeta_1) - y/n^2} + O_P(\sqrt{\rho_n}/n). \tag{A.21}$$

From the proof of Lemma 3.2, Equation A.13, and Equation A.21, we have: $\epsilon_1, \epsilon_2 < \frac{p+q}{2}$.

Also note that $\epsilon_1, \epsilon_2 = \Omega_P(-(p-q)\zeta_2^2 + \sqrt{\rho_n}/n)$. Hence, by the same argument as in Lemma 3.2, $|(p+q)/2 - \lambda| \leq \max(|\epsilon_1|, |\epsilon_2|) = \frac{p-q}{2} + O_P(1/n)$ by (A.21).

Finally we see that

$$t = \Theta(\frac{\epsilon_1 + \epsilon_2}{\rho_n}) = \Theta\left((p-q)\zeta_2^2/\rho_n\right).$$

In addition, condition (13) implies that $\zeta_2^2 = \Omega_P(1)$, we see that $t = \Omega_P(1)$ using (A.15).

Next, using (12) and A.19,

$$\kappa_1 + \kappa_2 = 4t\left(\frac{\mu_1 + \mu_2 - 1}{2}\left(\frac{p+q}{2} - \lambda\right) + \frac{(\mu_1 - \mu_2)(p-q)}{4} + O_P(\rho_n/\sqrt{n})\right),$$

$$\kappa_1 - \kappa_2 = 4t\left(\frac{\mu_1 + \mu_2 - 1}{2}\left(\frac{p+q}{2} - \lambda\right) - \frac{(\mu_1 - \mu_2)(p-q)}{4} + O_P(\rho_n/\sqrt{n})\right).$$

In (A.20), $b_i^{(0)}$ is of smaller order than the other terms and it suffices to consider $n(\kappa_1 + \sigma_i \kappa_2 + r_i^{(0)})$. Since $|r_i^{(0)}| = O_P\left(\sqrt{\frac{\rho_n \log^2 n}{n}}\right)$ (see proof of Theorem 3.4), for any pair $i \in C_1$ and $j \in C_2$ we have

$$(\kappa_1 + \kappa_2 + r_i^{(0)})(\kappa_1 - \kappa_2 + r_j^{(0)})$$

$$\leq (\kappa_1^2 - \kappa_2^2) + O\left(\max(|r_i^{(0)}|, |r_j^{(0)}|)\max(|\kappa_1|, |\kappa_2|)\right)$$

$$= (\kappa_1^2 - \kappa_2^2) + O_P\left((p-q)\sqrt{\frac{\rho_n \log^2 n}{n}}\right)$$

$$= t^2 (p-q)^2\left((\mu_1 + \mu_2 - 1)^2 - (\mu_1 - \mu_2)^2 + O_P\left(\frac{1}{p-q}\sqrt{\frac{\rho_n \log^2 n}{n}}\right)\right) < 0.$$

Thus $n(\kappa_1 + \kappa_2 + r_i^{(0)})$ and $n(\kappa_1 - \kappa_2 + r_j^{(0)})$, for $i, j$ in different blocks, have opposite signs. We will now check if $n(\kappa_1 + \sigma_i \kappa_2 + r_i^{(0)}) \to \infty$, and it suffices to lower bound $n(|\kappa_2| - |\kappa_1| - \max_i |r_i^{(0)}|)$. Since $|\mu_1 - \mu_2| \geq 2|\mu_1 + \mu_2 - 1| + O_P\left(\frac{\sqrt{\rho_n \log^2 n/n}}{p-q}\right)$,

$$n(|\kappa_2| - |\kappa_1| - \max_i |r_i^{(0)}|) \geq nt\left(|\mu_1 - \mu_2|(p-q) - |\mu_1 + \mu_2 - 1|(p-q) - O_P\left(\sqrt{\frac{\rho_n \log^2 n}{n}}\right)\right)$$

$$\geq nct(p-q)|\mu_1 - \mu_2| = \Theta\left(|\mu_1 - \mu_2|^3 n \frac{(p-q)^2}{\rho_n}\right),$$

for some constant $c$, so as long as $|\mu_1 - \mu_2| \geq \left( \frac{\rho_n \log n}{n(p-q)^2} \right)^{1/3}$.

Thus $\kappa_1 + \sigma_i \kappa_2 + r_i^{(0)}$ is growing to infinity with an order bounded below by $\Omega_P(\log n)$.

If $n(\kappa_1 + \kappa_2 + r_i^{(0)}) > 0$, since $\psi_i^{(1)} = g(n(\kappa_1 + \sigma_i \kappa_2) + b_i^{(0)} + n r_i^{(0)})$, we have $\psi^{(1)} = \mathbf{1}_{\mathcal{C}_1} + O_P(\exp(-\Omega(\log n)))$. The case $\kappa_1 + \kappa_2 + r_i^{(0)} < 0$ is similar. $\square$