[Reviews · NeurIPS 2018]

Reviewer 1



This paper studies the BCAVI variational method on the simplest blockmodel (2 classes occuring equally likely, and 2 parameters for within & between class densities). The main results seem to be: 1) The first main result is that if the density parameters are known, then the stationary points of the population BCAVI updates are derived. Note that the population BCAVI updates are not the same as the updates given noisy data, so there is a gap between the result and what BCAVI would actually do in practice. This is unusual for this type of work. The authors seem to suggest that the shortfall is merely trivial and that they didn't prove it because they weren't interested in that direction, but then it seems bad judgement not to prove it in the supplement, since it is of practical importance. To me, it is better to say nothing than to make an unsubstantiated claim (while of course a clearly stated conjecture is fine), so I am penalizing the paper somewhat for this. The authors also discuss how the result gives the volume of the basins of attraction -- but this seems again to be done informally, and without any concrete statements such as "the volume of the basin of attraction of such and such local optimum goes to X under such and such conditions". As a result, I think it will be hard for many readers (including me) to gauge some of the author's claims. 2) The second main result does deal with the actual BCAVI updates on noisy data (as opposed to the population updates), and removes the assumption that the density parameters are known. They show that if the class labels are initialized randomly then BCAVI converges to a noninformative local minimum. I believe this result is new. Previous results like [1] require a better initialization than random to show convergence, which complements this result in the paper. 3) A third result is that if the class labels are initialized to be correlated with the truth, then BCAVI recovers the true classes. It was unclear to me whether this result differs significantly from the result of [1], which also requires the initialization to be correlated with the truth. The paper is not as clearly written as [1], perhaps owing to the 8 page limit. The strength of the results is also less clear, particularly the 1st and 3rd results which may have shortfalls that were discussed above. With additional work to clarify and strengthen the results, the paper might be greatly improved. [1] Zhang A., Zhou, H. "Theoretical and Computational Guarantees of Mean Field Variational Inference For Community Detection" EDIT: while MC simulation is still unsatisfying to me, I acknowledge now that the basins of attraction (of the population iterates, not the actual ones) are characterized in a way that gives more insight into their nature. Readers would benefit if the discussion of differences from [1] was added to the paper, otherwise they may have trouble differentiating this results -- yours results deserve better discussion. Score has been raised from 6 to 7.

Reviewer 2



The authors present an analysis of coordinate ascent variational inference in the stochastic block model with two components. They show that if a pair of model parameters are known, then the majority of random initialization give the truth. However, when they aren't random restarting is pointless. Overall, I'm a big fan of this kind of analysis, it helps understand if we should trust results from approximation algorithm. I have several questions: - What happens when the number of communities exceeds 2? - Does the Gibbs sampler recover the truth, when p and q are unknown? Does it take exponential time?

Reviewer 3



Considers a simple variant of the stochastic block model (SBM) for networks with 2 communities/clusters. Previous work has studied the asymptotic behavior of the global optimizer of the mean field variational inference objective; this paper instead analyzes the dynamics of mean field coordinate ascent updates. It shows recovery of the correct community structure for most random initializations given correct model parameters, but for a particular way of initializing inference with unknown community interaction probabilities, inference converges to an uninformative local optimum with high probability. ORIGINALITY: Recent papers have given related analyses of inference algorithms for mixture models, but study of inference dynamics for the SBM appears novel. CLARITY: The fairly technical results here would clearly be easier to present in an extended journal paper. This submission puts some proofs in the supplement, but in my view this only partially succeeds in making the main manuscript readable. It would benefit from moving a few more details from Sec. 3 to the supplement, and adding additional explanation. There is a lot of mathematical notation, some introduced with no explanation even on first use, and I found myself repeatedly hunting through the paper to find the definition of terms. Some more detailed comments: 1) Sec. 2 does not mention that the reason probabilities B are constrained symmetric is that A is the symmetric adjacency matrix of an undirected graph. (Several previous papers have developed SBM variants for directed graphs, with non-symmetric B.) 2) Sec. 2 needs to be slower in explaining how the algorithm is changing in the last two equations. (Batch updates are not coordinate ascent on the mean field objective, and in general are not guaranteed convergent. The "population" variant justification is ignored.) Also \xi^(s) is not defined, and its connection to \psi^(s) is not specified. 3) Proposition 3.1 statement, second "eigenvalues" should be "eigenvectors". Alpha variables are an example of notation introduced without comment in the middle of a technical argument, that you then assume the reader will remember when trying to understand your experiments in Sec. 4. EXPERIMENTS: The (limited) experiments in Sec. 4 are a useful supplement to the earlier theory, but the text here is terse and includes insufficient explanation for the conceptual interpretation of the variables involved. In particular, a much clearer explanation for Figure 1(a) is needed. TECHNICAL CONTENT & SIGNIFICANCE: The technical derivations needed to establish the main theorems are non-trivial, and the key arguments in the main paper all seem correct. (I did not verify all proofs from the appendices.) Given that the model is restricted to two communities, it found it unsurprising to find (Theorem 3.3) that mean field converges to the correct community structure with known parameters. (This is similar to the positive results of [25] for 2-cluster Gaussian mixtures, but I would argue that the negative results in [13] with 3 or more clusters are more illuminating and important.) The negative results of Sec. 3.2 are more interesting. They essentially say that if inference is initialized with random community assignments, and model parameters are updated from there, the probability of identification of correct communities is exponentially small. However, I disagree with the claim in Sec. 5 that this is "typically done by most practitioner". The researchers who I know that do applied work with cluster-based models know that initializing with random partitions tends to lead to very poor local optima. For this SBM, here is one example of a more standard initialization: 1) Sample a "reasonable" set of initial block interaction probabilities. (For this model, this proposal would pick p >> q.) 2) Do at least one, and perhaps several, updates of the community probabilities psi given the fixed initial (p,q). 3) From this point, update (p,q) and then iterate as analyzed in the paper. The positive results in Sec. 3.1 with correct (p,q) suggest that if the (p,q) sampled in step #1 is within a neighborhood of the correct values, this algorithm may lead to a good optimum. Analysis of such a more realistic inference scheme could be an interesting area for future research.